# Effects of Water Temperature on Gonads Growth in *Ambystoma mexicanum* Axolotl Salamanders

**DOI:** 10.3390/ani13050874

**Published:** 2023-02-28

**Authors:** Chester R. Figiel

**Affiliations:** United States Fish and Wildlife Service, Warm Springs Fish Technology Center, Warm Springs, GA 31830, USA; chester_figiel@fws.gov

**Keywords:** Gonadosomatic Index, gametogenesis, climate change, amphibian

## Abstract

**Simple Summary:**

Amphibians are especially sensitive to temperature due to their lack of thermal regulation. Aquatic temperatures influence growth and development during the larval period and may impact sexual development. This study examined the rearing of axolotl salamanders from juvenile to adulthood at four temperatures to examine how this factor influenced reproduction output. Rearing temperature affected egg and testis production and indicates that males and females should be reared at different temperatures to optimize reproductive output. Additionally, axolotl salamanders may be especially sensitive to high temperatures suggesting that the reintroduction and translocation of axolotls into former and new habitats should identify water temperature as an important component in conservation efforts.

**Abstract:**

The thermal environment is a major factor influencing amphibians. For example, amphibian reproduction occurs in specific temperatures, and minor changes in this aspect could have negative impacts on this biological process. Understanding the potential effects of temperature on reproductive output is important from both an ecological and captive breeding colony point of view. I examined temperature effects on reproduction in axolotl reared from egg to adult at 4 temperatures (15 °C, 19 °C, 23 °C, and 27 °C) These adult axolotls (*n* = 174) were measured and weighed, dissected, and their gonads were removed and weighed to obtain an individual’s reproductive allocation. Female axolotls reared at 23 °C had a greater Gonadosomatic Index (GSI) compared to axolotl reared at each of the other temperatures with axolotls reared at 27 °C having the lowest reproductive output. Moreover, all GSI pair-wise comparisons in the four temperature treatments were significantly different from each other (ANOVA, F (3, 66) = 61.681, *p* < 0.0001). Additionally male rearing temperature significantly influenced GSI (ANOVA, F (3, 89) = 10.441, *p* < 0.0001). Male axolotls reared at 19 °C had significantly greater GSI compared to males reared at the three other temperatures. There were no statistical differences among each of the other pair-wise comparisons. As seen in this experiment, axolotls may be especially sensitive to climate-driven warming due to their highly permeable skin and paedomorphic life history. Understanding how axolotls and other amphibian species adjust to the challenges of climate change is important in the management of this imperiled taxa.

## 1. Introduction

Caudates are especially sensitive to changing temperatures due to their lack of thermal regulation. Aquatic temperatures influence growth and development during the larval period and may impact sexual development and induce sex reversal [1,2,3]. Two literature reviews [4,5] examined the influence of temperature on the developmental plasticity of amphibian larvae and determined that age and size at the onset of metamorphosis were generally lowest at the warmest temperatures. Amphibian larvae that develop faster and metamorphose at a smaller body potentially impact fitness–related traits [6,7]. Further, larval amphibians may be more vulnerable than adults to temperature variations and are likely to encounter higher variation in temperature living in the aquatic environment. Larval amphibians often prefer specific temperatures in lab experiments and this behavior can change depending on developmental stage [8,9]. Temperature preference also occurs in adult obligatory aquatic salamanders (*Amphiuma tridactylum*) [10], and acclimation temperature can influence larval and adult preferred temperature. For example, aquatic salamanders, *Cryptobranchus alleganiensis* (hellbenders) and *Necturus maculosus* (mudpuppies), selected the lower temperatures in a gradient when acclimated at 5 °C and avoided warmer temperatures [11]. In hellbenders, water temperature can impact the pH and P_CO2_ [12], and in *Ambystoma texanum* (small-mouth salamanders), water temperature can influence physiological immune function [13], and metabolic rates [14], Rearing temperature can reduce growth [15] and survival of *Andrias davidianus* (the Chinese giant salamander) [16] and play an important role in the initiation of reproductive cycles [17], with low temperatures inducing early timing of female gamete maturation, as shown in Siberian salamanders (*Salamandrella keyserlingii*) captured during breeding migrations [18]. Thus, seasonal temperatures may affect spermatogenesis and ovulation in temperate salamander species and their endocrine e and exocrine control and constraints [19,20].

Notwithstanding these studies, little is known whether and how water temperature affects gonads growth in urodeles. This information is of importance for the development of temperature regimes for rearing amphibians in captive colonies (including germplasm repository development), and in understanding the effects of climate induced changes on life histories traits including reproduction.

The present study examines the effect of water temperature on the Gonadosomatic Index (GSI) in the Axolotl, *Ambystoma mexicanum*. These salamanders are imperiled and near extinction in nature because of invasive predatory fish, pollution, and urbanization [21,22,23,24]. Axolotls are thriving in domestication—as exotic pets, or in laboratory colonies where they are used as model organisms for biological research. This species has been cultured in the lab as surrogates for wild *Ambystoma* species on the research of cold storage and cryopreservation of spermatophores [25,26] and for examining temperature influence on reproductive physiology [27]. Axolotls possess features typical of salamander larvae, including external gills, and are paedomorphic (i.e., retaining juvenile characteristics as adults), which allows them to exploit water throughout their entire lifespan. By living in water, however, the species may be more vulnerable to changes in water temperature or quality compared to amphibians that are both land and water dwelling. Thus, it is important to know how the effects of temperature can influence reproduction of this species and other cold-blooded vertebrates.

## 2. Materials and Methods

### 2.1. Animals

This study took place at the Warm Springs Fish Technology Center, U. S. Fish and Wildlife Service, in Warm Springs, Georgia. Male and female adult axolotl originally obtained from the University of Kentucky’s Ambystoma Genetic Stock Center (AGSC) were paired together in 20 °C water in three containers to obtain eggs for this experiment. Approximately 75 larvae (three weeks old) from each of three clutches were mixed and divided into four groups and put in four tanks (244 cm × 61 cm × 24 cm) that were filled with artesian spring water (pH x = 6.8 ± 0.2) and recirculated through a chiller/heater which maintained temperatures at 15 °C, 19 °C, 23 °C, and 27 °C. Temperatures were checked using a YSI Pro 20i digital meter every other day and thermostat was adjusted if needed. Rearing temperatures were within the range of temperatures (10 °C to 28 °C) that axolotl have been successfully cultured [28]. Axolotls were exposed to a 12 h light and 12 h dark photoperiod and fed brine shrimp daily until larvae were approximately 5–6 cm in total length and thereafter fed ad libitum rations of a prepared pellet diet obtained from the AGSC. This feed was broadcast haphazardly throughout the tank 5–6 days a week and larvae ate freely. Periodically, the four systems were tested for ammonia (x < 0.1), nitrite (x = 0.05), and alkalinity (x = 15 mg/L).

### 2.2. Gonads Collection

As axolotls matured in culture tanks, individuals were sampled starting at 12 months of age (starting in March 2022)—those not sampled remained exposed to culture temperatures until collected at a later date (up to 15.5 month of age). When collected, individuals were euthanized by adding 10.0 g of Tricaine Methane-Sulfonate (MS-222) and two parts of sodium bicarbonate to 1 L of water. Salamanders were placed in the solution and 10 min after visible breathing stopped, Snout-Vent-Length (SVL, distance from the tip of the snout to the anterior angle of the vent)) (mm) was measured, then individuals were dried with a paper towel, and weighed (body mass: BM) to within 0.1 g using an electronic balance. Individuals were dissected, testis or egg masses removed, and these were weighed to 0.1 g and photographed. All axolotls were subsequently frozen individually.

### 2.3. Statistical Analysis 

Gonadosomatic index (GSI) was obtained by the dividing testis or eggs mass versus the body mass (in grams). GSI is widely used in studies of reproductive cycles of different groups of animals [29,30,31,32]. These data were arcsine square root transformed to minimize the heterogeneity of variances among treatments [33]. Body size data were log-transformed before analysis. An ANOVA was performed on these data using the Excel2016.lnk software package (Microsoft, Redmond Washington, DC, USA) to analyze the effect of rearing temperature on the GSI (*p* = 0.05 level of significance). If the *p*-value from the ANOVA was less than the significance level, a Tukey–Kramer post hoc test was done to examine which groups are different from each other.

## 3. Results

### 3.1. Females

Female body size was significantly affected by rearing temperature. Axolotls reared at 15 °C and 23 °C and had a greater SVL and body mass compared to axolotls reared at 19 °C and 27 °C temperatures (Table 1). These body size differences were statistically different. SVL: ((ANOVA, F (3, 65) = 13.625, *p* < 0.0001), Body mass: (ANOVA, F (3, 65) = 31.907, *p* < 0.0001).

Additionally, female axolotls reared at 23 °C had a greater GSI compared to axolotl reared at each of the other temperatures (Figure 1). All pair-wise comparisons in GSI in the four temperature treatments were significantly different from each other (ANOVA, F (3, 66) = 61.681, *p* < 0.0001, Figure 1), with axolotls reared at 27 C having the lowest reproductive output.

### 3.2. Males

Male body size was significantly affected by rearing temperature. Axolotls reared at 23 °C had a greater SVL (115.4 mm) and body mass (93.4 g) compared to axolotls reared at other temperatures (Table 1). These body size differences were statistically different. SVL: ((ANOVA, F (3, 86) = 4.572, *p* = 0.0052), Body mass: (ANOVA, F (3, 86) = 14.143, *p* < 0.0001).

Rearing temperature significantly influenced GSI (ANOVA, F (3, 89) = 10.441, *p* < 0.0001). Male axolotls reared at 19 °C had significantly greater GSI compared to males reared at the three other temperatures (Figure 2). Males in the warmest water temperature (27 °C) had the smallest GSI compared to axolotls reared at 15 °C and 23 °C temperatures, but these differences were not statistically different. 

## 4. Discussion

### 4.1. Females

Body size of females and the growth development of gonads depended strongly on temperature regime. Axolotls were larger both in length and weight when reared at 15 and 23 °C. Interestingly, gonads size of axolotls reared at the coldest temperature (15 °C) did not increase in proportion to body mass. These salamanders may have the capacity to allocate food energy to growth rather than to reproduction as temperature conditions at 15 °C may not be ideal. Alternatively, if temperature environments are optimal, females will growth larger and produce more eggs. This likely occurred in females reared at 23 °C as axolotls were larger and the GSI was significantly greater compared to females at the three other temperatures.

### 4.2. Males

Similar to females, male body size was greatest at 15 °C and 23 °C, however males reared at 19 °C had a significantly greater GSI compared to males reared at the three other temperatures. Again, the potential for an energy trade-off between growth and reproduction is likely for males reared at different temperatures. 

### 4.3. Effects of High Rearing Temperature and Climate Warming

Both females and males were smaller in body size and produced relatively a smaller percentage of gonads when reared at 27 °C. This was particularly true for females. High temperatures may influence not only gonads production but may also affect gamete maturation or increase gamete mortality as seen in fish and mammals [34,35]. Seasonal temperature variation can influence spermatogenesis and ovulation is some salamander species [36,37].

I4 nature, axolotl growth and reproduction could be affected by increased water temperatures that arise from climate change. This species originated in cold water (<16 °C) lakes in Mexico and ideal water temperatures for rearing axolotls in the lab is 16–18 °C. Water temperatures above 24 °C can put stress on the animals and result in a loss of appetite (personal observation) and may make axolotls more vulnerable to parasites. A previous study [38] suggested that maintaining axolotl at 24 °C or below can accelerate growth without serious detrimental effects but rearing temperatures above that would reduce axolotl developmental rates. This occurred in the present experiment as rearing temperatures of 27 °C resulted in axolotl reduced body size and decreased the relative mass of testis and the number of eggs. It is possible that the 27 °C water temperature affected hormonal function in these salamanders and suppressed gametogenesis. Future studies are needed to address the role of temperature on hormones, gamete quality, and paternal investment in offspring. 

## 5. Conclusions

These results suggest that the reintroduction and translocation of axolotls into former and new habitats should identify water temperature as an important component in conservation efforts. Previous research that examined axolotl distribution found this species in water temperatures ranging between 16 to 23 °C [24]. As seen in this experiment, males and females should be reared at different temperatures to optimize reproductive output and may be especially sensitive to high temperatures. Whether temperature influences other maternal contribution to offspring, i.e., input into other fitness-related qualities (e.g., egg or hatching size) is presently unknown. 

While this study examined the effects of constant temperatures on axolotl reproduction, additional research should focus on the consequences of variable thermal environments. The ability of species to adapt to this extrinsic factor behaviorally or physiologically is little known. Understanding how axolotls and other amphibian species adjust to the challenges of temperature variation is important in the management of this imperiled taxa.

## Figures and Tables

**Figure 1 animals-13-00874-f001:**
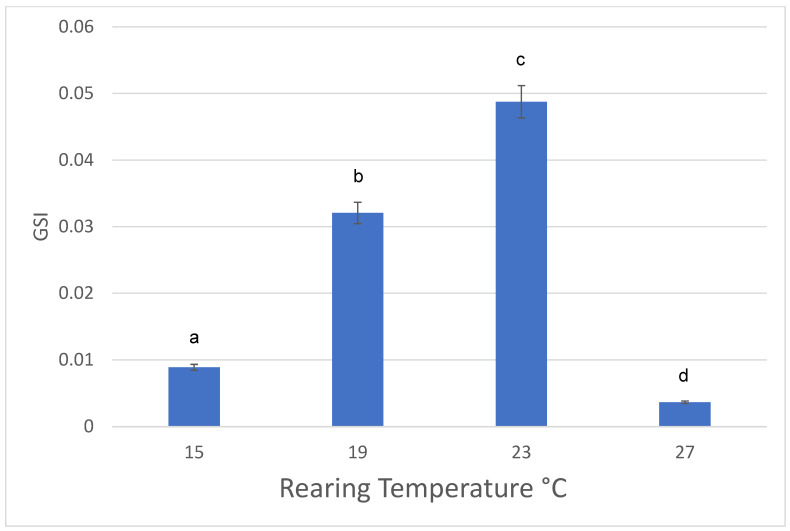
Effects of rearing temperature (15 °C, 19 °C, 23 °C, and 27 °C) on Gonadosomatic Index (GSI) of female axolotls, *Ambystoma mexicanum*. Different letters indicate statistically significant differences among temperatures. Mean ± standard deviation.

**Figure 2 animals-13-00874-f002:**
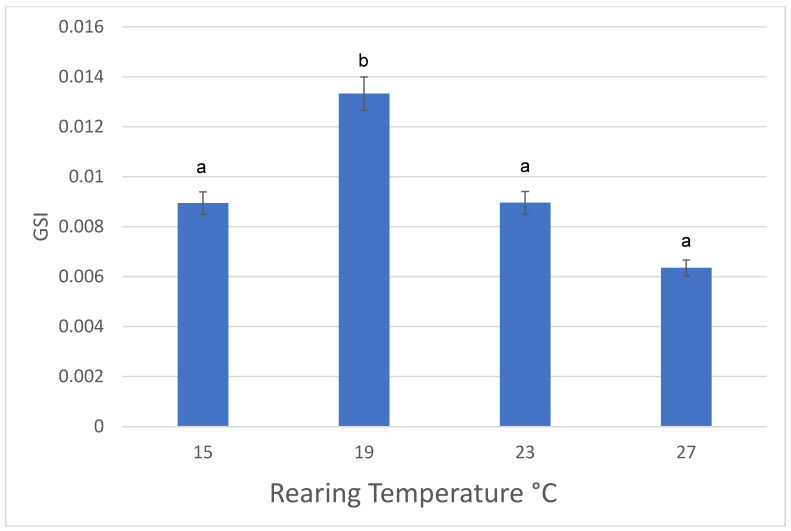
Effects of rearing temperature (15 °C, 19 °C, 23 °C, and 27 °C) on Gonadosomatic Index (GSI) of male axolotls, *Ambystoma mexicanum*. Different letters indicate statistically significant differences among temperatures. Mean ± standard deviation.

**Table 1 animals-13-00874-t001:** Effects of rearing temperature (15 °C, 19 °C, 23 °C, and 27 °C) on snout-vent-length (SVL) (mm) and body mass (g) of female and male axolotls, *Ambystoma mexicanum*. Different letters indicate statistically significant differences among temperatures. Mean ± standard deviation.

**Females**	**15 °C**	**19 °C**	**23 °C**	**27 °C**
*n*	25	21	16	21
SVL	110.6 ± 12.6 ^a^	103.8 ± 5.4 ^b^	114.4 ± 5.4 ^a^	101.6 ± 8.9 ^b^
Body Mass	95.9 ± 12.6 ^a^	70.6 ± 7.9 ^b^	93.1 ± 12.7 ^a^	57.0 ± 18.3 ^b^
**Males**	**15 °C**	**19 °C**	**23 °C**	**27 °C**
*n*	27	30	20	12
SVL	111.1 ± 4.1 ^a^	109.5 ± 5.6 ^a^	115.0 ± 6.1 ^b^	109.2 ± 8.6 ^a^
Body Mass	86.5 ± 7.8 ^a^	72.3 ± 12.3 ^a^	92.1 ± 17.8 ^b^	72.1 ± 13.1 ^a^

## Data Availability

The data presented in this study are available upon request from the corresponding author. Additionally, data files have been uploaded to FigShare (doi: 10.6084/m9.figshare.21713945.

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
