# Peer review of "Effects of Water Temperature on Gonads Growth in Ambystoma mexicanum Axolotl Salamanders"

_animals, 2023, doi:10.3390/ani13050874_

Round 1
Reviewer 1 Report
The paper is well written, the experimental design is appropriate, the methods are adequately described and the conclusions are sound. I suggest some revision in the text, there are some mispelling (The author's name, the title and the citation at Page 1). Besides, there are no reference to ethical approvance of animal treatments by local authorities, if they are not required it should be clearly stated. In conclusion, I suggest to accept the paper with minor revisions.
Author Response
Thank you to this reviewer as their comments improved the manuscript.
Spell check has been used and has corrected all misspelling.
Ethical approval has been added to the text.
Reviewer 2 Report
The Authors have examined the potential effects of temperature on reproductive output of female and male Axolotl Salamander, Ambystoma mexicanum. Even if the topic of paper is important for an ecological and reproductive point of view, some little corrections/integrations in the text can improve the final version of paper.
1) the presentation of graphics (histograms) can be improved and also must be added the significance on them.
2) in the introduction section it should be inserted some sentences on particular characteristics of this species and also cited an important review on Amphibians (RK Rastogi et al., 2005, Hormonal regulation of reproductive cycles in amphibians. Amphibian biology, Volume 6, pag. 2045-2177, Eds. Surrey Beatty & Sons Sydney).
3) please avoid to insert Axolotl Salamanders as first author of paper....
Author Response
Thank you to this reviewer as their comments improved the manuscript in the following ways.
1) I have included the significance in the Table legend to clarify which treatment means were statistically different from each other.
2. I have added characteristics of this species in the introduction and cited Rastogi et al., 2005, as well as Rastogi et al., 2011.
3. I have made the correction regarding first author
Reviewer 3 Report
The paper deal with rearing temperature of a salamander specie. Although the paper is quite simple as it did not explore any physiological parameter and only performed some metrics of the body size and gonad size at a final time point of development, the culture conditions of this specie had not been studied in deep. This data might be important to define the optimal culture conditions to improve other studies as regeneration, development or embryogenesis studies already being performed in this specie.
Anyway, author should review the subsequent comments:
It is not clear the time of sampling. How many times did the specimens kewp in the experimental temperature before sampling? Where all the individuals collected at the same time point?
Why are not the statistic reported in the table 1. Were not statisticall differences between groups regarding the SVL and mass data? Would be easier for readers to understand the data whether the statistically differences would be indicated using asterisk or letters the statistical differences on the table than only in the text. To know differences between groups authors might use a post-hoc test before the ANOVA test.
Author Response
Thank you to this reviewer as their comments improved the manuscript in the following ways.
I have added to the method section to clarify the time of sampling began at 12 months of age and continued until 15.5 months of age. Also that all individuals were maintained in their appropriate temperature treatment.
In the Table 1 legend, I included the statistical differences including the Tukey-Kramer post-hoc test used to determine which groups were different from each other.